# Walking on the Fiber:
# A Simple Geometric Approximation for Bayesian Neural Networks

**Alfredo Reichlin**                                    *alfrei@kth.se*
*KTH Royal Institute of Technology*

**Miguel Vasco**                                        *miguelsv@kth.se*
*KTH Royal Institute of Technology*

**Danica Kragic**                                       *dani@kth.se*
*KTH Royal Institute of Technology*

**Reviewed on OpenReview:** *https://openreview.net/forum?id=NsuPykrjOd*

## Abstract

Bayesian Neural Networks provide a principled framework for uncertainty quantification by modeling the posterior distribution of network parameters. However, exact posterior inference is computationally intractable, and widely used approximations like the Laplace method struggle with scalability and posterior accuracy in modern deep networks. In this work, we revisit sampling techniques for posterior exploration, proposing a simple variation tailored to efficiently sample from the posterior in over-parameterized networks by leveraging the low-dimensional structure of loss minima. Building on this, we introduce a model that learns a deformation of the parameter space, enabling rapid posterior sampling without requiring iterative methods. Empirical results demonstrate that our approach achieves competitive posterior approximations with improved scalability compared to recent refinement techniques. These contributions provide a practical alternative for Bayesian inference in deep learning.

## 1 Introduction

Bayesian Neural Networks (BNNs) have emerged as a principled framework for uncertainty quantification in deep learning by treating model parameters as random variables and inferring their posterior distribution. This Bayesian perspective is crucial for tasks requiring reliable decision-making under uncertainty, including medical diagnosis (Leibig et al., 2017), autonomous driving (Feng et al., 2018), and weather forecasting (Cofino et al., 2002). Despite their appeal, the high dimensionality and non-linearity of modern neural network parameter spaces render exact posterior inference intractable, requiring the development of efficient approximation techniques.

A widely used method to make a deterministic, already-trained neural network Bayesian is the Laplace approximation, which estimates the posterior distribution of the parameters as a Gaussian centered at the Maximum a Posteriori (MAP) estimate (MacKay, 1992; Daxberger et al., 2021a). This approach is particularly convenient due to its simplicity and post-hoc applicability. However, the Laplace approximation faces significant challenges in modern deep networks. The computation and inversion of the Hessian scale poorly with network size (Martens, 2010), and its reliance on local curvature limits its ability to capture the complex, non-linear geometry of the posterior in over-parameterized models (Fort & Jastrzebski, 2019; Bergamin et al., 2024).

In contrast, sampling techniques, though less frequently used in recent years, can be surprisingly effective in exploring the posterior when the parameter space of loss minima resides on a much lower-dimensional

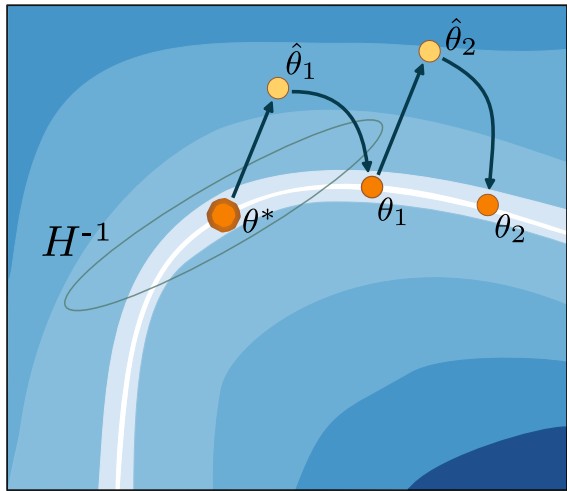

Figure 1: We propose a novel sampling scheme to approximate the posterior of a trained network. When the loss landscape of a trained network over its parameters (blue regions) is low on a very low-dimensional curve (white region), a simple Hessian ($H^{-1}$) fails in capturing its distribution. Our sampling scheme is composed of two iterative steps: randomly perturb a given solution (yellow points) then refine to minimize the loss function over the given dataset (orange points). This proposed scheme allows to estimate posteriors of arbitrary shapes and it is well-suited when the space of solutions in a network is much lower dimensional than the parameter space.

manifold than the full parameter space (Garipov et al., 2018). By leveraging this property, sampling methods can efficiently explore the posterior with fewer samples, especially in over-parameterized settings. However, relying on sampling for posterior estimation can still be computationally expensive if every new inference requires a fresh sampling process. These limitations highlight the need for a method that combines the efficiency of sampling-based exploration with a more scalable and flexible posterior representation.

In this work, we propose a variation of a sampling method to explore the geometry of the loss landscape in the parameter space of neural networks. Our approach perturbs parameters around the MAP estimate using a set of drift directions and refines them with gradient updates, effectively maintaining computational efficiency as the network size increases. Using the data collected from our sampling process, we introduce a novel objective function to learn a structured latent representation of the posterior by deforming the original parameter space. This learned representation enables the direct generation of new parameter samples without relying on computationally expensive iterative sampling or restrictive variational inference approximations. We refer to these two components collectively as **MetricBNN**. Empirically, we demonstrate the efficacy of our method in estimating uncertainty from trained models in both regression and classification tasks. In particular, our approach produces better-calibrated uncertainty estimates than Gaussian approximations, especially on real-world datasets and high-dimensional tasks.

## 2 Related Work

**Bayesian Neural Networks.** BNNs provide a principled approach to uncertainty quantification by treating the network parameters $\theta$ as random variables and modeling their posterior distribution $p(\theta \mid \mathcal{D})$. However, exact inference is intractable due to the high dimensionality of parameter spaces in modern neural networks. Various methods have been proposed to approximate the posterior. Variational Inference (VI) approximates the posterior by optimizing a surrogate distribution, often in the form of a mean-field Gaussian (Graves, 2011; Blundell et al., 2015). Extensions to hierarchical and amortized variational methods have improved scalability but often struggle to capture complex posterior structures (Zhang et al., 2018). The Laplace Approximation (LA) defines the posterior locally around the MAP estimate using a Gaussian distribution (MacKay, 1998; Daxberger et al., 2021a). Despite its simplicity and efficiency, it is limited by its reliance on local curvature

and the Gaussian assumption. Sampling-based methods such as Stochastic Gradient Langevin Dynamics (SGLD) (Welling & Teh, 2011) and Hamiltonian Monte Carlo (HMC) (Neal, 2011) generate posterior samples by simulating stochastic dynamics. While these methods are more flexible, their computational cost often makes them impractical for large-scale neural networks.

**Improving Posterior Samples.** To address these limitations, recent works have sought to refine and extend the Laplace approximation by introducing more expressive approximate posteriors. Kristiadi et al. (2022) combines Laplace approximation with normalizing flows for a non-Gaussian posterior, refining the base Gaussian distribution. Similarly, Immer et al. (2021) refines Laplace approximation by leveraging Gaussian variational methods and Gaussian processes, improving linearized Laplace posterior accuracy. Miller et al. (2017) iteratively builds a mixture model to improve the posterior approximation by adding components to the variational distribution. Eschenhagen et al. (2021) combines multiple pre-trained models to form a weighted sum of Laplace approximations, improving posterior flexibility. Havasi et al. (2021) introduced auxiliary variables to locally refine mean-field variational approximations, achieving better fit in regions of interest. Bergamin et al. (2024) extends the Laplace approximation by leveraging Riemannian geometry to model the posterior distribution on a manifold, improving accuracy for non-linear loss landscapes.

In this paper, we re-evaluate the efficiency of sampling methods for posterior estimation, particularly in the context of over-parameterized neural networks. Through empirical results, we demonstrate that our proposed simple sampling framework can outperform modern posterior refinement techniques in efficiency and accuracy. Furthermore, our novel use of an autoencoder moves beyond the Gaussian formulation of the posterior, enabling a flexible, easy-to-sample representation that significantly improves performance. This approach allows us to capture the non-linear geometry of the posterior while maintaining computational simplicity, addressing key limitations of both classical and modern refinement methods. Differently from classic hypernetwork approaches, MetricBNN learns a structured latent representation of near-optimal parameter regions explicitly for posterior approximation. This enables us to generate parameter samples efficiently while preserving a geometry-informed posterior structure, directly targeting calibrated Bayesian uncertainty estimation rather than task-conditioned adaptation.

## 3 Preliminaries

BNNs provide a principled framework for quantifying uncertainty in neural network predictions by treating the model parameters as random variables. This section introduces the relevant background on BNNs, discusses the Laplace approximation for posterior estimation, and highlights its limitations.

Consider an independent and identically distributed (i.i.d.) dataset $\mathcal{D} = \{(x_i, y_i)\}_{i=1}^N$ where $x_i \in \mathbb{R}^D$ and $y_i \in \mathbb{R}^C$. Let $f_\theta : \mathbb{R}^D \to \mathbb{R}^C$ denote a parametric function (e.g., a neural network) with parameters $\theta \in \mathbb{R}^K$. The goal is to model the predictive distribution: $p(y' \mid x', \mathcal{D}) = \int p(y' \mid x', \theta) p(\theta \mid \mathcal{D}) \, d\theta$, where $p(\theta \mid \mathcal{D})$ is the posterior distribution of the parameters given the dataset. Bayesian inference provides a framework for estimating the posterior $p(\theta \mid \mathcal{D})$ using Bayes' theorem: $p(\theta \mid \mathcal{D}) = \frac{p(\mathcal{D}|\theta)p(\theta)}{p(\mathcal{D})}$, where $p(\mathcal{D} \mid \theta)$ is the likelihood of the data, $p(\theta)$ is the prior distribution over the parameters, and $p(\mathcal{D})$ is the marginal likelihood or evidence. In practice, computing the evidence $p(\mathcal{D})$ is often intractable, making direct posterior computation challenging.

### 3.1 Laplace Approximation

The Laplace approximation is a classic method for approximating the posterior distribution $p(\theta \mid \mathcal{D})$ using a Gaussian centered at the MAP estimate. Let $\mathcal{L}(\theta)$ denote the regularized negative log-likelihood:

$$\mathcal{L}(\theta) = -\log p(\mathcal{D} \mid \theta) - \alpha \log p(\theta), \tag{1}$$

where $\alpha$ is a regularization coefficient and depends on the choice of prior for the parameters. Assuming this to be the Normal distribution, the regularization can be rewritten as an Euclidean norm.

One of the main advantages of the Laplace approximation is that it allows to define a posterior distribution given an already fully trained network, *post-hoc*. When given a fully trained network we assume access to

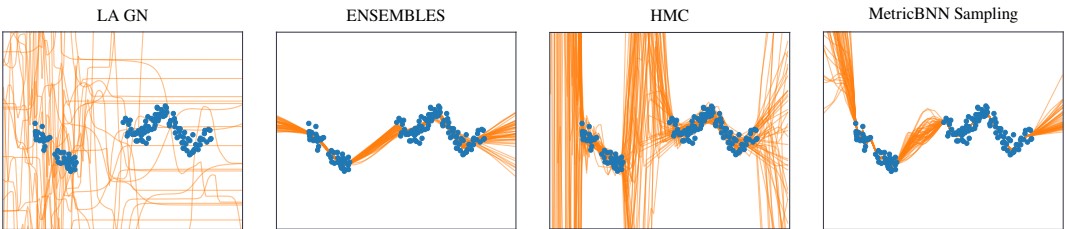

Figure 2: Posterior samples for Regression task. The blue points represent the dataset, the orange lines are samples from the estimated posteriors. Our proposed sampling method correctly captures the uncertainty in the data gap.

the set of parameters, $\theta^*$, that minimize the loss function of Equation 1. This is the MAP solution. The exponential of this loss function is proportional to the posterior distribution as $p(\theta \mid \mathcal{D}) \propto p(\mathcal{D} \mid \theta)p(\theta)$. Laplace approximation methods propose to approximate the posterior distribution with a second-order Taylor expansion around the MAP of the loss function. This results in a Gaussian approximation of the parameters with the mean being the MAP solution and the covariance being the inverse of the Hessian computed in the MAP. The posterior can then be defined as:

$$q(\theta) = \mathcal{N}(\theta^*, H^{-1}), \tag{2}$$

where $H = \nabla^2 \mathcal{L}(\theta^*)$ is the Hessian of the loss function at $\theta^*$.

### 3.2 Limitations of the Laplace Approximation

While widely used, the Laplace approximation has several limitations:

1. **Local Approximation:** It captures only the local curvature of the loss landscape around $\theta^*$, ignoring the broader structure of the posterior, which can be highly non-Gaussian in high-dimensional spaces (Wilson & Izmailov, 2020).

2. **Scalability:** Computing and inverting the Hessian is computationally expensive for large-scale neural networks, limiting its applicability to small models (Martens, 2010).

3. **Positive-Definiteness:** In over-parameterized networks, the Hessian is often not positive definite, making it difficult to define a valid Gaussian approximation. Regularization or approximate methods are sometimes used to mitigate this issue, but these approaches can introduce biases (Daxberger et al., 2021a).

The limitations of the Laplace approximation motivate the need for alternative methods that better capture the true posterior distribution. Specifically, the posterior for BNNs often lies on a complex, non-linear manifold in parameter space, which the Laplace approximation fails to represent. This motivates our approach, which combines efficient exploration of the parameter space with a flexible latent representation to better approximate the posterior.

## 4 Method

To address the limitations of the Laplace approximation and improve posterior estimation in BNNs, we propose *MetricBNN*, a two-step framework. The first step involves locally exploring the parameter space around the MAP estimate using a simple sampling method. The second step learns a latent representation to construct a flexible posterior distribution that captures the complex geometry of the parameter space.

### 4.1 Exploring Neighbor Solutions

Given a trained neural network with parameters $\theta^*$ that minimize the regularized loss function (Equation 1), the parameter space of deep networks is known to exhibit a high degree of redundancy due to overparameterization

Sampling Distribution    MetricBNN Distribution    SVD Approximation

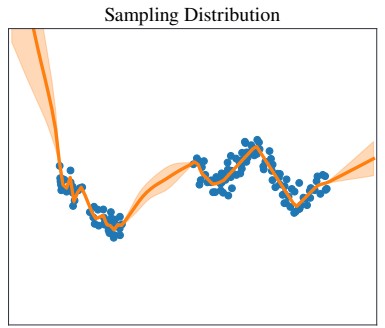 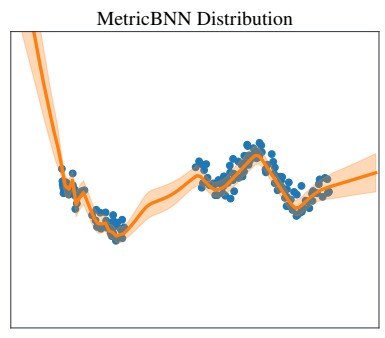 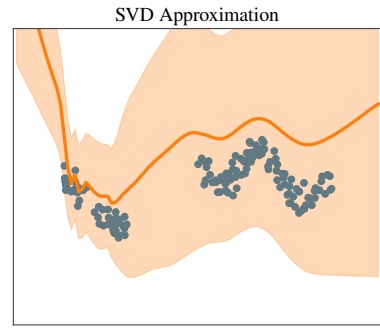

Figure 3: Estimated posterior for the regression task. The blue points represent the dataset and in orange the mean and standard deviation of posteriors sampled from the estimated distributions. A naive SVD approximation of the solutions found with the sampling scheme fails in correctly representing the true posterior. Our proposed MetricBNN posterior correctly approximates it.

and reparametrization invariances Fort & Jastrzebski (2019); Garipov et al. (2018). These properties create a connected, lower-dimensional manifold of near-optimal solutions surrounding the MAP estimate $\theta^*$. This phenomenon, often referred to as the *linear connectivity assumption*, suggests that different sets of parameters can achieve similar performance, even when interpolating between them (Garipov et al., 2018; Fort & Jastrzebski, 2019; Brea et al., 2019).

In particular, empirical studies on the loss landscapes of neural networks have shown that minima are often connected by low-loss paths, forming smooth and structured regions in the parameter space (Garipov et al., 2018). This implies that the posterior distribution is not confined to a single mode but instead spans a broader, non-linear manifold. Capturing this geometric complexity is crucial for accurately modeling the posterior. By exploring neighboring solutions around $\theta^*$, we aim to gather representative samples that reflect the underlying structure of this manifold, providing a richer and more accurate approximation of the posterior distribution.

Estimating the posterior distribution of the parameters amounts to identifying the distribution of these solutions. To achieve this, we propose collecting a set of such solutions using the following sampling technique (MetricBNN sampling):

1. **Initialization:** Start with $N$ particles, each initialized at the MAP estimate $\theta^*$, i.e., $\theta_{i,0}$.

2. **Random Drift Assignment:** Assign each particle a random drift vector $d_i$, sampled as a unit vector with uniform orientation in the parameter space.

3. **Iterative Exploration:** For $T$ time steps, update each particle's position as:

   (a) **Drift:** Perturb the particle by adding the corresponding random drift.

   $$\hat{\theta}_{i,t+1}^0 = \theta_{i,t} + \alpha \cdot d_i. \tag{3}$$

   (b) **Refinement:** After each drift update, refine the particle's position using $M$ steps of gradient descent to ensure alignment with the loss landscape.

   $$\theta_{i,t+1}^{m+1} = \hat{\theta}_{i,t+1}^m - \eta \nabla_\theta \mathcal{L}(\hat{\theta}_{i,t+1}^m) \tag{4}$$

   $$\theta_{i,t+1} = \hat{\theta}_{i,t+1}^M \tag{5}$$

This procedure generates a set of $N \times T$ viable solutions near $\theta^*$. These samples represent a local exploration of the posterior distribution of interest, capturing the diversity of solutions around the MAP estimate. Figure 2(on the right) illustrates an example of these solutions for a two-dimensional regression problem.

While sampling methods have traditionally been considered computationally expensive for high-dimensional neural networks, leading to their relative neglect in favor of variational inference and other approximation

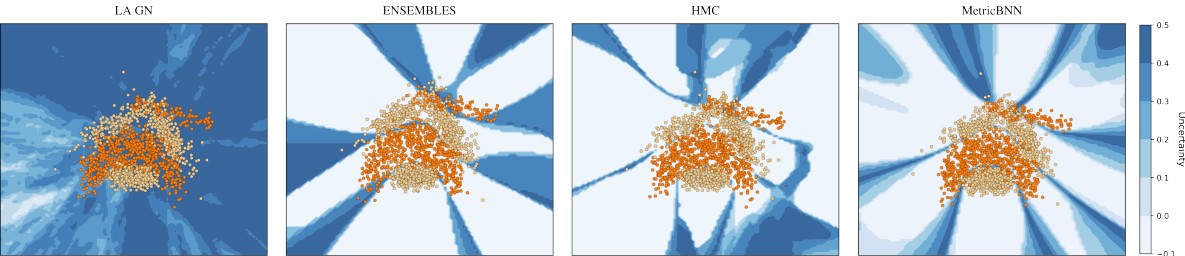

Figure 4: Estimated posterior for the Banana classification task. The orange and yellow points represent the data of the two classes in the classification dataset. In blue is the value of the uncertainty of the posteriors sampled from the estimated distributions.

techniques (MacKay, 1998; Blundell et al., 2015), we argue that these concerns warrant re-evaluation in light of the linear connectivity assumption. Specifically, the high redundancy and structured geometry of neural network parameter spaces suggest that meaningful posterior exploration can be achieved through efficient sampling on a lower-dimensional, connected manifold of solutions (Garipov et al., 2018; Fort & Jastrzebski, 2019). This implies that the computational complexity of sampling methods may scale more favorably with the dimensionality of modern networks than previously assumed.

The empirical covariance matrix of the collected solutions provides an improved local Gaussian approximation of the posterior distribution. However, due to the extreme non-linearity of the solution manifold, a Gaussian approximation alone fails to accurately capture the true structure of the posterior distribution. Addressing this limitation requires moving beyond the Gaussian assumption.

## 4.2 Defining a Posterior

While the empirical covariance of the collected samples provides a basic Gaussian approximation, it fails to capture the non-linear geometry of the posterior, see Figure 3. Addressing this limitation requires moving beyond the Gaussian assumption. Given that the drift applied in the Drift step (3a) is small enough, we can assume that every element in between two sets of parameters is a viable solution. The curve defined by all the samples can, however, be arbitrarily non-linear. To overcome this, we propose to learn a representation that maps the parameters into a structured latent space where the collected trajectories of parameters are linearly distributed.

We define a latent representation through an autoencoder framework:

- $\varphi : \Theta \to Z$: An encoder that maps neural network parameters $\theta$ to a latent space $Z \subseteq \mathbb{R}^k$.

- $\varphi^{-1} : Z \to \Theta$: A decoder that reconstructs parameters $\theta$ from the latent space.

The training dataset for the autoencoder consists of the collected samples, i.e., $D_\theta$. The goal is to learn a latent space where the posterior distribution is well-structured and linearly interpolable. We achieve this by optimizing the following loss function:

$$\mathcal{L} = \lambda_+ \mathcal{L}_+ + \lambda_- \mathcal{L}_- + \mathcal{L}_d, \tag{6}$$

with:

$$\mathcal{L}_+ = \mathbb{E}_{D_\theta} \left[ \left( ||\varphi(\theta) - \varphi(\theta')|| - \frac{1}{T} \right)^2 \right], \tag{7}$$

$$\mathcal{L}_- = \mathbb{E}_{D_\theta} \left[ -\log \left( ||\varphi(\theta) - \varphi(\theta'')|| \right) \right], \tag{8}$$

$$\mathcal{L}_d = \mathbb{E}_{D_\theta} \left[ ||\varphi^{-1}(\varphi(\theta)) - \theta||^2 \right], \tag{9}$$

where $\theta$ and $\theta'$ are two sets of parameters of the same trajectory and successive time step while $\theta''$ is another randomly sampled set of parameters. Both $\lambda_+$ and $\lambda_-$ are scalar values. The role of these terms is as follows:

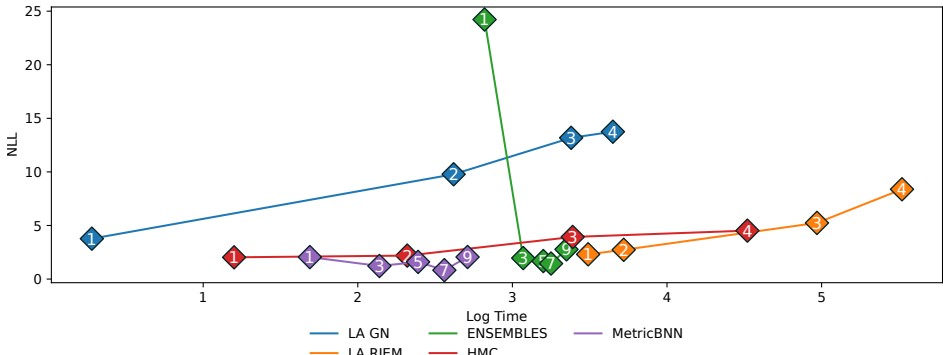

Figure 5: Trade-off between network size and computational complexity on the regression task. Each marker shows NLL (lower is better) for a network with the indicated number of hidden layers. MetricBNN's cost grows only mildly with depth.

- $\mathcal{L}_+$: encourages local distances between successive samples are preserved.

- $\mathcal{L}_-$: encourages these sets of parameters to stretch in the learned latent space by maximizing every pair-wise distance.

- $\mathcal{L}_d$: encourages reconstruction, mapping latent representations back to their original parameter space.

Using the trained autoencoder, we define the posterior distribution in the latent space $Z$, MetricBNN posterior. Each trajectory of samples is treated as independent, with a uniform probability assigned to each trajectory. Along each trajectory, we place a uniform probability between the MAP solution and the last sample of such trajectory, i.e., $\theta_{i,T}$. Sampling from the posterior involves: Sample a trajectory index $i$ uniformly, Sample a scaling factor $\epsilon \sim \text{Uniform}(0,1)$. Compute the parameter sample:

$$\theta = \varphi^{-1}\left(\varphi(\theta^*) + \epsilon \cdot \left(\varphi(\theta_{i,T}) - \varphi(\theta^*)\right)\right). \tag{10}$$

This method captures the broader geometry of the posterior while maintaining computational efficiency. We provide a pseudo-code of the whole method in Appendix B

## 5 Experiments

In this section, we evaluate the ability of our proposed method to approximate the posterior efficiently. We first assess the quality of our sampling technique in capturing model uncertainty using simple regression and classification tasks. We then analyze the computational complexity of our approach as the network size increases. Finally, we present quantitative results on real-world datasets, evaluating the negative log-likelihood (NLL) and the expected calibration error (ECE) of our posterior distribution.

We benchmark our method against Bayesian inference baselines that collectively span local–Gaussian, stochastic–sampling, and deterministic–ensembling paradigms.

Gauss–Newton Laplace (LA GN) (Daxberger et al., 2021a) fits a Gaussian around the MAP solution using the Gauss–Newton curvature, offering a lightweight post-hoc posterior of any trained network. For some experiments, we also include a low-rank estimate of the Laplace approximation (LA LowRank) and a linearized version (Lin LA).

Riemannian Laplace (Riem LA) (Bergamin et al., 2024) extends this idea by replacing the Euclidean metric with an appropriate Riemannian version, yielding a curvature-aware Gaussian that better respects the manifold structure of the parameter space. We also include the linearized version of this approach (Riem Lin LA).

For fully stochastic references we include Hamiltonian Monte Carlo (HMC) (Neal, 2011) and its scalable stochastic-gradient variant (SGLD) (Welling & Teh, 2011), both of which provide asymptotically exact posterior samples at the cost of increased computation.

Complementing these, Deep Ensembles (Lakshminarayanan et al., 2017) (ENSEMBLES) approximate the posterior with a set of independently trained networks, while (SWAG) (Maddox et al., 2019) captures the first two moments of SGD iterates to form a low-rank Gaussian that can be sampled cheaply.

Finally, (ESPRO) (Benton et al., 2021) exploits mode-connecting simplices to generate functionally diverse checkpoints without extra training epochs, yielding a fast ensembling baseline.

We conduct experiments across five different settings:

**Simple Regression:** We evaluate our method on the one-dimensional regression problem introduced in Snelson & Ghahramani (2005). The dataset consists of 200 points, with 50 held out to assess the model's ability to estimate uncertainty. We use a fully connected neural network with three hidden layers of 32 units and ReLU activations.

**Simple Classification:** We test our method on the Banana dataset, a two-dimensional binary classification task with 5,300 points, of which 30% are reserved for testing. We use a fully connected network with two hidden layers of 6 units and Tanh activations.

**Classification on UCI Datasets:** To evaluate performance on real-world structured data, we experiment with six classification datasets from the UCI repository (Markelle et al., 2023). We use a fully connected network with two hidden layers of 32 units and ReLU activations.

**Image Classification:** For high-dimensional problems, we experiment with the MNIST (LeCun, 1998), FashionMNIST (Xiao et al., 2017) and CIFAR10 (Krizhevsky, 2009) datasets. For MNIST and FMNIST we use a shallow convolutional neural network with two convolutional layers followed by three fully connected layers with Tanh activations. For CIFAR10 we instead use a ResNet18 architecture (He et al., 2016) and compute the posterior on the last two layers of the network.

**OOD Classification:** We test the ability of the model in classifying MNIST images with an increasing amount of rotations (Daxberger et al., 2021b) and CIFAR10 images with 15 different realistic noises (Hendrycks & Dietterich, 2019).

## 5.1 Scalability of the Proposed Sampling Method

Approximating the posterior distribution in neural networks is challenging due to the high-dimensional and non-linear nature of the parameter space. The standard Laplace approximation has been shown to struggle in these settings (Ritter et al., 2018; Lawrence, 2001), as it assumes a Gaussian posterior, which may not align well with the actual distribution.

Our proposed sampling method provides a more flexible alternative, allowing for a tighter and better-calibrated posterior that is independent of the loss landscape's curvature. Figure 2 compares our method with the Laplace approximation on the toy regression task, demonstrating a significantly improved uncertainty estimation.

Beyond accuracy, scalability is a key factor in Bayesian inference for deep networks. Many posterior approximation techniques rely on computing second-order derivatives, making them computationally expensive as the network size grows. To assess the computational trade-off, we analyze the inference time and NLL performance as the number of layers in the network increases. Figure 5 presents these results for our method, LA GN, Riem LA, ENSEMBLES, HMC. Results show that Hessian-based methods and HMC scale poorly with size. ENSEMBLES have a higher temporal cost as they require the training of multiple models. Moreover, they suffer from a very low-dimensional network as they converge to the same solution and thus result in low variance solutions. Our sampling approach maintains a reasonable computational cost while preserving accuracy. Furthermore, our learned latent posterior model enables rapid parameter sampling, further reducing the overhead compared to iterative sampling.

Table 1: Quantitative results on the UCI datasets, including NLL, lower is better, and ECE, lower is better.

|  | Model | Australian | Breast | Glass | Ionosphere | Vehicle | Waveform |
|---|---|---|---|---|---|---|---|
| NLL | LA GN | $0.71 \pm 0.06$ | $0.73 \pm 0.07$ | $2.28 \pm 0.28$ | $0.72 \pm 0.09$ | $0.8 \pm 0.16$ | $0.88 \pm 0.09$ |
|  | LA LowRank | $0.69 \pm 0.03$ | $0.64 \pm 0.10$ | $2.06 \pm 0.15$ | $1.08 \pm 0.21$ | $0.76 \pm 0.05$ | $0.96 \pm 0.04$ |
|  | Riem LA | $0.54 \pm 0.07$ | $0.59 \pm 0.1$ | $1.42 \pm 0.21$ | $0.17 \pm 0.02$ | $0.65 \pm 0.02$ | $0.3 \pm 0.01$ |
|  | Lin LA | $0.69 \pm 0.04$ | $0.61 \pm 0.04$ | $3.59 \pm 1.17$ | $0.42 \pm 0.04$ | $0.68 \pm 0.01$ | $0.4 \pm 0.02$ |
|  | Riem Lin LA | $0.74 \pm 0.05$ | $2.32 \pm 0.74$ | $15.92 \pm 0.05$ | $13.5 \pm 0.66$ | $0.65 \pm 0.01$ | $0.38 \pm 0.02$ |
|  | ENSEMBLES | $0.51 \pm 0.07$ | $0.62 \pm 0.10$ | $0.90 \pm 0.08$ | $0.25 \pm 0.05$ | $0.63 \pm 0.00$ | $0.30 \pm 0.01$ |
|  | HMC | $1.30 \pm 0.31$ | $1.52 \pm 0.61$ | $1.49 \pm 0.40$ | $0.80 \pm 0.30$ | $0.65 \pm 0.02$ | $0.43 \pm 0.03$ |
|  | SGLD | $0.64 \pm 0.16$ | $0.65 \pm 0.10$ | $1.36 \pm 0.03$ | $0.26 \pm 0.07$ | $0.73 \pm 0.02$ | $0.72 \pm 0.09$ |
|  | ESPRO | $0.76 \pm 0.25$ | $0.63 \pm 0.12$ | $0.94 \pm 0.17$ | $0.22 \pm 0.10$ | $0.64 \pm 0.03$ | $0.32 \pm 0.01$ |
|  | SWAG | $0.70 \pm 0.04$ | $0.59 \pm 0.06$ | $1.56 \pm 0.08$ | $0.25 \pm 0.08$ | $0.68 \pm 0.00$ | $0.30 \pm 0.01$ |
|  | MetricBNN | $0.66 \pm 0.05$ | $0.57 \pm 0.04$ | $1.07 \pm 0.07$ | $0.17 \pm 0.05$ | $0.69 \pm 0.02$ | $0.3 \pm 0.01$ |
| ECE | LA GN | $13.61 \pm 4.34$ | $18.53 \pm 9.51$ | $12.25 \pm 3.56$ | $28.03 \pm 7.23$ | $21.36 \pm 9.26$ | $21.68 \pm 6.40$ |
|  | LA LowRank | $17.12 \pm 3.17$ | $16.66 \pm 9.26$ | $17.72 \pm 10.25$ | $49.68 \pm 5.82$ | $18.88 \pm 6.29$ | $13.82 \pm 4.38$ |
|  | Riem LA | $10.28 \pm 4.83$ | $13.60 \pm 2.91$ | $21.30 \pm 7.77$ | $5.57 \pm 0.78$ | $5.80 \pm 1.10$ | $5.62 \pm 0.70$ |
|  | Lin LA | $10.62 \pm 1.71$ | $18.45 \pm 4.57$ | $15.32 \pm 7.21$ | $28.80 \pm 1.67$ | $7.31 \pm 1.30$ | $23.00 \pm 1.82$ |
|  | Riem Lin LA | $8.47 \pm 2.71$ | $20.90 \pm 2.58$ | $14.10 \pm 2.14$ | $6.84 \pm 2.31$ | $10.75 \pm 1.88$ | $2.51 \pm 0.72$ |
|  | ENSEMBLES | $8.09 \pm 1.37$ | $15.46 \pm 3.77$ | $13.47 \pm 1.57$ | $8.24 \pm 1.14$ | $2.90 \pm 1.30$ | $1.39 \pm 0.26$ |
|  | HMC | $16.03 \pm 2.87$ | $26.04 \pm 4.98$ | $18.32 \pm 2.43$ | $9.24 \pm 2.10$ | $8.46 \pm 1.17$ | $6.53 \pm 0.96$ |
|  | SGLD | $13.38 \pm 3.36$ | $15.12 \pm 6.35$ | $18.40 \pm 3.09$ | $8.12 \pm 2.58$ | $12.12 \pm 3.03$ | $8.70 \pm 4.52$ |
|  | ESPRO | $11.62 \pm 3.33$ | $15.41 \pm 3.37$ | $14.03 \pm 2.91$ | $6.47 \pm 1.72$ | $8.29 \pm 1.62$ | $2.51 \pm 0.96$ |
|  | SWAG | $9.80 \pm 6.35$ | $14.29 \pm 2.49$ | $3.87 \pm 2.75$ | $7.44 \pm 2.12$ | $1.18 \pm 0.98$ | $2.39 \pm 0.59$ |
|  | MetricBNN | $27.33 \pm 8.08$ | $12.42 \pm 4.38$ | $11.51 \pm 5.46$ | $5.74 \pm 0.97$ | $10.18 \pm 10.20$ | $1.50 \pm 0.53$ |

Table 2: Quantitative results on image datasets, including NLL, lower is better, and ECE, lower is better.

|  | Model | MNIST | FMNIST | CIFAR10 |
|---|---|---|---|---|
| NLL | LA GN | $2.42 \pm 0.03$ | $2.24 \pm 0.08$ | $2.29 \pm 0.09$ |
|  | LA LowRank | $1.35 \pm 0.04$ | $1.67 \pm 0.06$ | $2.19 \pm 0.08$ |
|  | Riem LA | $1.11 \pm 0.11$ | $1.20 \pm 0.02$ | $0.84 \pm 0.03$ |
|  | Lin LA | $0.96 \pm 0.08$ | $1.27 \pm 0.13$ | $1.93 \pm 0.12$ |
|  | Riem Lin LA | $0.62 \pm 0.09$ | $0.87 \pm 0.07$ | $0.83 \pm 0.06$ |
|  | ENSEMBLES | $0.17 \pm 0.00$ | $0.47 \pm 0.00$ | $1.55 \pm 0.02$ |
|  | SGLD | $1.60 \pm 0.20$ | $1.98 \pm 0.26$ | $0.78 \pm 0.06$ |
|  | ESPRO | $0.15 \pm 0.02$ | $0.51 \pm 0.02$ | $0.84 \pm 0.05$ |
|  | SWAG | $0.24 \pm 0.01$ | $0.89 \pm 0.11$ | $1.97 \pm 0.33$ |
|  | MetricBNN | $0.12 \pm 0.01$ | $0.58 \pm 0.09$ | $0.70 \pm 0.04$ |
| ECE | LA GN | $4.78 \pm 2.97$ | $12.31 \pm 2.39$ | $4.59 \pm 1.88$ |
|  | LA LowRank | $50.58 \pm 3.20$ | $18.70 \pm 3.07$ | $9.74 \pm 5.82$ |
|  | Riem LA | $47.14 \pm 9.19$ | $26.78 \pm 7.83$ | $33.91 \pm 8.23$ |
|  | Lin LA | $29.43 \pm 1.32$ | $19.63 \pm 1.18$ | $35.46 \pm 12.67$ |
|  | Riem Lin LA | $21.20 \pm 2.00$ | $15.19 \pm 1.92$ | $32.10 \pm 3.15$ |
|  | ENSEMBLES | $10.52 \pm 0.13$ | $11.33 \pm 0.29$ | $8.94 \pm 0.17$ |
|  | SGLD | $13.83 \pm 7.47$ | $14.55 \pm 6.06$ | $7.45 \pm 1.25$ |
|  | ESPRO | $1.56 \pm 0.20$ | $4.36 \pm 0.79$ | $10.40 \pm 0.33$ |
|  | SWAG | $13.47 \pm 1.07$ | $25.25 \pm 4.95$ | $3.47 \pm 2.76$ |
|  | MBNN | $1.57 \pm 0.31$ | $13.57 \pm 6.89$ | $9.26 \pm 0.45$ |

## 5.2 Quality of the Proposed Geometric Posterior

A potential concern is whether the posterior obtained via our sampling method could be approximated using a naive covariance computation, similar to the Laplace approximation. However, as shown in Figure 3, this approach still does not yield accurate uncertainty estimates. By contrast, our latent posterior representation

effectively captures the true posterior structure, demonstrating the benefits of learning a structured parameter space.

### 5.3 Performance on UCI Regression and Image Classification

We further evaluate our approach to real-world structured data by testing it on standard UCI regression benchmarks. Table 1 reports NLL and ECE results, showing that our method remains competitive with the baselines. In Appendix we provide details on the computational cost as well demonstrating that our method offers a good balance between accuracy and computational cost.

To assess performance in high-dimensional settings, we train convolutional neural networks on MNIST and FashionMNIST and CIFAR10. Table 2 presents NLL and ECE results.

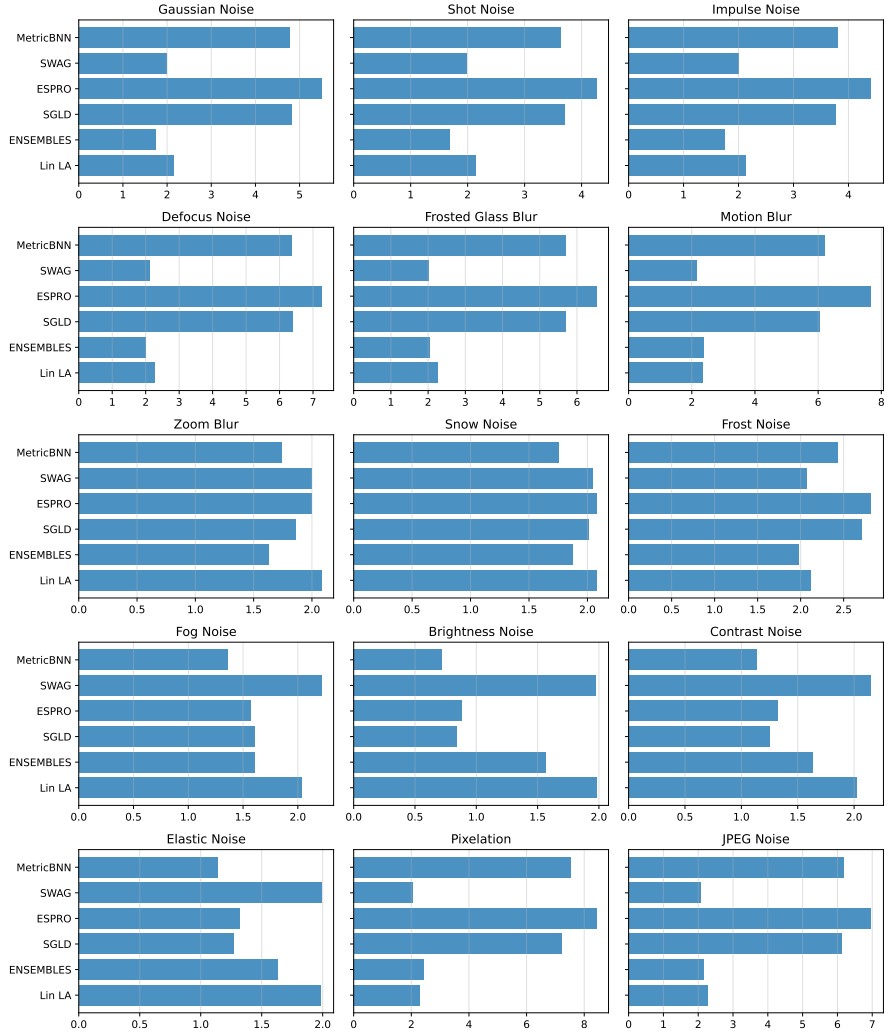

Figure 6: Robustness to semantic and photometric shift CIFAR10. Each panel reports test NLL across each corruption.

### 5.4 OOD Classification

We evaluate all methods under two canonical OOD settings. For every corruption we report the log-likelihood. All networks are first trained on clean data only. Posterior approximations are then fit *post-hoc*.

**Geometric shift:** progressively rotating MNIST test images by 0°–90° (Daxberger et al., 2021b). Figure 7 shows that MetricBNN has competitive log-likelihood performances for rotations up to 30 degrees. Performances drop when the rotations become more severe as distinguishing between the numbers is not always possible.

**Semantic + photometric shift:** applying the fifteen corruption families of CIFAR10 (Hendrycks & Dietterich, 2019). Figure 6 shows competitive performances for MetricBNN for some of the perturbations.

Our experiments highlight three key takeaways: Our sampling-based posterior exploration provides a more flexible and well-calibrated uncertainty estimate, particularly as network size increases. The proposed latent posterior model enables efficient posterior sampling, avoiding the computational overhead of iterative methods. Our approach remains competitive in terms of NLL and ECE on real-world datasets while being significantly more computationally efficient.

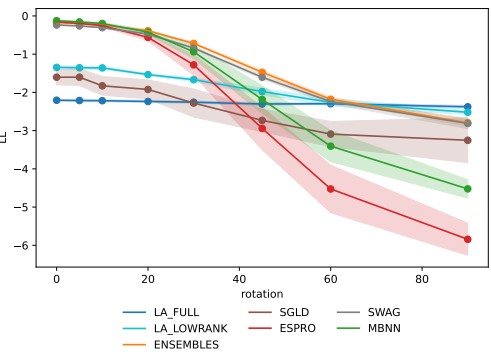

Figure 7: Robustness to geometric shift (Rotated MNIST). Test NLL and standard deviations of MNIST image classification versus rotation angle; lower is better.

## 6 Discussion

In this work, we revisited scalable Bayesian inference in deep learning, benchmarking our method, MetricBNN, against a broad range of baselines including Laplace approximations, variational methods, deep ensembles, and sampling-based techniques. Across structured and high-dimensional tasks, MetricBNN consistently achieves competitive or superior negative log-likelihood and calibration while maintaining efficiency as network size increases. We observe that while Laplace-based methods and their geometric extensions can perform well in certain settings, their performance is notably sensitive to the parameter count, network capacity, and the quality of the MAP solution used for centering the approximation. Differences in MAP training conditions and model size can therefore lead to the lower scores observed in our experiments compared to those reported in other Laplace-focused studies. Our sampling-based exploration combined with a learned latent posterior helps mitigate these sensitivities, capturing non-Gaussian, non-linear posterior structures more robustly across diverse regimes.

Across our experiments, MetricBNN demonstrates strong empirical performance on structured datasets, high-dimensional image classification, and out-of-distribution (OOD) robustness benchmarks. On structured UCI datasets, MetricBNN achieves negative log-likelihood and calibration scores competitive with or better than Laplace (LA GN, Riem LA) and variational baselines, while providing a more flexible posterior representation. On high-dimensional datasets like MNIST, FashionMNIST, and CIFAR10, MetricBNN maintains low NLL and ECE scores, matching or outperforming deep ensembles and methods such as SWAG, while avoiding the computational overhead of training multiple models or computing Hessians at scale. In OOD settings, MetricBNN shows consistent robustness under geometric shifts (rotated MNIST) and photometric and semantic corruptions (CIFAR10), providing well-calibrated uncertainty estimates across perturbation levels.

Compared to fully sampling-based methods such as Hamiltonian Monte Carlo (HMC) and SGLD, MetricBNN achieves comparable uncertainty estimates while being significantly more computationally efficient, making it practical for large-scale neural network applications. In contrast to deep ensembles, which require training multiple networks to achieve uncertainty calibration, MetricBNN provides comparable calibration with a single network, improving efficiency and scalability. Compared to variational inference methods, which often rely on restrictive mean-field approximations, MetricBNN captures the non-Gaussian structure of the posterior by leveraging the geometry of the parameter space through structured latent representations.

Finally, while hypernetworks typically generate weights conditioned on input or task information for adaptation, our approach fundamentally differs by using an autoencoder to learn a structured latent space of near-optimal parameter regions for scalable posterior sampling. This allows MetricBNN to efficiently generate parameter

samples that reflect the geometry of the loss landscape without the use of expensive iterative techniques or variational methods.

## 7    Conclusions

In this work, we proposed a simple variation of sampling-based techniques tailored to explore the posterior geometry of Bayesian Neural Networks efficiently, even in over-parameterized settings. By leveraging the low-dimensional structure of loss minima, our method achieves competitive posterior approximations while maintaining scalability as network size increases. Additionally, we introduced a model that learns a deformation of the parameter space based on the collected samples, enabling rapid posterior sampling without requiring iterative methods. Our empirical results demonstrate that this approach improves posterior accuracy and computational efficiency compared to recent refinement techniques. These contributions provide a practical and flexible framework for Bayesian inference in deep learning, offering new directions for scalable uncertainty quantification in complex models.

**Acknowledgments**

This work was supported by the Swedish Research Council (2019-00748), the Knut and Alice Wallenberg Foundation, the European Research Council (ERC-BIRD-884807) and the European Horizon 2020 CANOPIES project.

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

# A Experimental Details

This section provides additional details regarding the experimental setup, including the models, hyper-parameters, and computational settings.

## A.1 Experimental Setup

We evaluate our method across four types of tasks: toy regression, toy classification, structured classification from the UCI repository, high-dimensional image classification and OOD classification.

**Toy Regression.** We consider the *Snelson 1D regression dataset* (Snelson & Ghahramani, 2005), consisting of 200 points, with 50 held out for evaluating uncertainty estimation. We use a fully connected neural network with three hidden layers of 32 units and ReLU activations. The model is trained using the Adam optimizer with a learning rate of $1 \times 10^{-3}$, batch size of 200, and L2 regularization of 0.01 for 50,000 epochs.

For the sampling procedure of our approach, we used a drift-scaling factor of $\alpha = 0.1$ and performed $T = 100$ sampling steps. At each step, we updated the parameters using $M = 10$ gradient refinement steps to improve posterior estimates. We initialized $N = 10$ particles, each following a perturbed trajectory in the parameter space, with a drift step scaled by an inner learning rate of $\eta = 0.001$. For the autoencoder-based posterior model, we projected the sampled parameter trajectories into a structured latent space of $k = 32$ dimensions. The autoencoder was trained using batch size 1024 for 1000 epochs, optimizing the contrastive loss with weight coefficients $\lambda_+ = 1.0$ and $\lambda_- = 1.0$. The architecture consists of an encoder with two fully connected hidden layers of 256 units and ReLU activations, and a decoder with a symmetric structure of two hidden layers with 256 units and ReLU activations. Training was performed using the Adam optimizer.

**Toy Classification.** For classification, we use the *Banana dataset*, a 2D binary classification problem with 5300 points, of which 30% are used for testing. The network consists of two hidden layers with 16 units and Tanh activations. The training procedure follows the regression setup, except with a smaller batch size (32) and 2500 training epochs. For our model, we use the same parameters as in the regression task.

**Structured Data (UCI Datasets).** To evaluate our method in structured classification settings, we test on six datasets from the *UCI repository* (Markelle et al., 2023), using a fully connected architecture with two hidden layers of 32 units and ReLU activations. The training follows the same schedule as before, with a batch size of 32 and 1000 training epochs. Our method is estimated with the same parameters as in the classification experiment except for $T = 50$ steps.

**High-Dimensional Image Classification.** For large-scale experiments, we train Bayesian neural networks on *MNIST* (LeCun, 1998), *FashionMNIST* (Xiao et al., 2017) and CIFAR10 (Krizhevsky, 2009). For MNIST and FAMNIST we use a shallow convolutional neural network consisting of two convolutional layers, followed by three fully connected layers, with Tanh activations. For CIFAR10 we use a standard ResNet18 architecture. Training follows the same setup as UCI experiments, but with a reduced number of epochs (100). Our method follows the same parameters as in the UCI experiment.

**OOD Classification.** We use the same networks described above for both MNIST and CIFAR10 as well as the same parameters for our method.

## A.2 Scalability and Computational Complexity

To conduct the experiment on the computational complexity depicted in Figure 5 we used a fully connected network with hidden layers from 1 to 9. Each layer with 64 units and ReLu as activation function. For each architecture, we measure the inference time and negative log-likelihood (NLL) performance. To compute the posterior of each model we use 100 samples of the parameters. All experiments were conducted on an NVIDIA RTX3080 GPU.

## B  Pseudocode

---

**Algorithm 1** Proposed Posterior Approximation Method

---

**Require:** Trained neural network $f_\theta$, dataset $\mathcal{D}$, drift scaling $\alpha$, sampling steps $T$, gradient steps $M$, particles $N$, inner learning rate $\eta$, autoencoder latent dimension $k$, training epochs $E$

**Ensure:** Approximate posterior $q(\theta)$

1: **Phase 1: Parameter Sampling**
2: Initialize $N$ particles at $\theta^*$ (MAP estimate)
3: **for** each particle $i = 1, \ldots, N$ **do**
4:      Sample random drift direction $d_i \leftarrow \frac{v}{||v||}, \quad v \sim \mathcal{N}(0, I)$
5:      **for** each step $t = 1, \ldots, T$ **do**
6:          Apply drift: $\theta_{i,t} = \theta_{i,t-1} + \alpha d_i$
7:          **for** each gradient step $m = 1, \ldots, M$ **do**
8:              Refine using gradient descent:
9:                  $\theta_{i,t} \leftarrow \theta_{i,t} - \eta \nabla_\theta \mathcal{L}(\theta_{i,t})$
10:          **end for**
11:      **end for**
12: **end for**
13: Store all sampled parameters $\Theta = \{\theta_{i,t}\}$
14: **Phase 2: Learning Latent Posterior Representation**
15: Train an autoencoder $\varphi : \Theta \to Z$ using:
16: **for** epoch $e = 1, \ldots, E$ **do**
17:      Sample minibatch of subsequent parameters $\theta, \theta' \sim \Theta$
18:      Compute latent embeddings: $z = \varphi(\theta)$ and $z' = \varphi(\theta')$
19:      Compute negative parameters by reshuffling the batch: $z''$
20:      Compute contrastive loss:
21:        $\mathcal{L}_+ = (||z - z'|| - 1/T)^2$
22:        $\mathcal{L}_- = -\log(||z - z''||)$
23:        $\mathcal{L}_d = ||\varphi^{-1}(z) - \theta||$
24:      Update autoencoder using $\mathcal{L} = \lambda_+ \mathcal{L}_+ + \lambda_- \mathcal{L}_- + \lambda_d \mathcal{L}_d$
25: **end for**
26: **Posterior Approximation and Sampling**
27: Sample trajectory index $j \sim \text{Categorical}(\{z_{\max,j}\}_{j=1}^N)$
28: Sample interpolation factor $\epsilon \sim \mathcal{U}(0, 1)$
29: Compute latent posterior sample: $z = z_{\text{MAP}} + \epsilon(z_{\max,j} - z_{\text{MAP}})$
30: Map to parameter space: $\theta = \varphi^{-1}(z)$
31: **return** Posterior sample $\theta$

---

## C Time Complexity Results

We provide a more detailed comparison of the computational complexity and the accuracy of our method and the baselines for the UCI experiment and the images experiment.

Table 3: Logarithmic time complexity in seconds and accuracy for the UCI datasets.

| | Model | Australian | Breast | Glass | Ionosphere | Vehicle | Waveform |
|---|---|---|---|---|---|---|---|
| Log Time | LA GN | $1.88 \pm 0.98$ | $1.54 \pm 0.8$ | $1.56 \pm 0.67$ | $1.88 \pm 0.72$ | $3.05 \pm 1.95$ | $2.68 \pm 0.48$ |
| | LA LowRank | $2.0 \pm 0.22$ | $1.24 \pm 0.28$ | $0.98 \pm 0.37$ | $1.47 \pm 0.48$ | $4.69 \pm 0.1$ | $3.68 \pm 0.16$ |
| | Riem LA | $4.36 \pm 4.06$ | $2.42 \pm 1.61$ | $3.69 \pm 3.63$ | $2.38 \pm 1.61$ | $3.33 \pm 3.0$ | $3.26 \pm 2.45$ |
| | Lin LA | $1.86 \pm 1.0$ | $1.52 \pm 0.4$ | $1.55 \pm 0.48$ | $1.89 \pm 0.65$ | $3.05 \pm 1.95$ | $2.68 \pm 0.31$ |
| | Riem Lin LA | $1.9 \pm 1.04$ | $1.63 \pm 0.6$ | $1.67 \pm 0.54$ | $1.93 \pm 0.65$ | $3.08 \pm 1.42$ | $2.69 \pm 1.53$ |
| | ENSEMBLES | $2.5 \pm 0.02$ | $2.14 \pm 0.03$ | $2.01 \pm 0.02$ | $2.2 \pm 0.02$ | $3.76 \pm 0.01$ | $3.32 \pm 0.02$ |
| | HMC | $3.3 \pm 0.2$ | $0.54 \pm 0.21$ | $1.73 \pm 0.09$ | $0.36 \pm 0.24$ | $2.62 \pm 0.42$ | $1.43 \pm 0.16$ |
| | SGLD | $1.38 \pm 0.03$ | $1.06 \pm 0.07$ | $0.95 \pm 0.05$ | $1.11 \pm 0.04$ | $2.58 \pm 0.03$ | $2.16 \pm 0.03$ |
| | ESPRO | $0.61 \pm 0.1$ | $0.33 \pm 0.16$ | $0.19 \pm 0.22$ | $0.31 \pm 0.18$ | $1.66 \pm 0.01$ | $1.27 \pm 0.03$ |
| | SWAG | $0.95 \pm 0.01$ | $0.62 \pm 0.07$ | $0.51 \pm 0.08$ | $0.68 \pm 0.08$ | $2.16 \pm 0.02$ | $1.72 \pm 0.01$ |
| | MetricBNN | $1.66 \pm 0.34$ | $1.61 \pm 0.29$ | $1.64 \pm 0.32$ | $1.76 \pm 0.35$ | $1.8 \pm 0.36$ | $1.82 \pm 0.41$ |
| Accuracy | LA GN | $0.49 \pm 0.12$ | $0.57 \pm 0.21$ | $0.17 \pm 0.11$ | $0.36 \pm 0.09$ | $0.45 \pm 0.04$ | $0.57 \pm 0.14$ |
| | LA LowRank | $0.55 \pm 0.05$ | $0.60 \pm 0.23$ | $0.12 \pm 0.06$ | $0.33 \pm 0.06$ | $0.46 \pm 0.04$ | $0.57 \pm 0.05$ |
| | Riem LA | $0.57 \pm 0.04$ | $0.72 \pm 0.01$ | $0.38 \pm 0.01$ | $0.93 \pm 0.02$ | $0.61 \pm 0.01$ | $0.86 \pm 0.00$ |
| | Lin LA | $0.56 \pm 0.06$ | $0.62 \pm 0.08$ | $0.11 \pm 0.09$ | $0.91 \pm 0.00$ | $0.60 \pm 0.01$ | $0.84 \pm 0.01$ |
| | Riem Lin LA | $0.75 \pm 0.01$ | $0.68 \pm 0.03$ | $0.48 \pm 0.18$ | $0.91 \pm 0.03$ | $0.58 \pm 0.02$ | $0.86 \pm 0.01$ |
| | ENSEMBLES | $0.82 \pm 0.02$ | $0.71 \pm 0.04$ | $0.72 \pm 0.04$ | $0.92 \pm 0.02$ | $0.72 \pm 0.00$ | $0.83 \pm 0.00$ |
| | SGLD | $0.73 \pm 0.03$ | $0.69 \pm 0.01$ | $0.52 \pm 0.08$ | $0.92 \pm 0.02$ | $0.50 \pm 0.02$ | $0.64 \pm 0.03$ |
| | ESPRO | $0.82 \pm 0.02$ | $0.77 \pm 0.01$ | $0.67 \pm 0.08$ | $0.96 \pm 0.01$ | $0.70 \pm 0.00$ | $0.86 \pm 0.01$ |
| | SWAG | $0.57 \pm 0.10$ | $0.71 \pm 0.04$ | $0.34 \pm 0.05$ | $0.92 \pm 0.02$ | $0.58 \pm 0.01$ | $0.86 \pm 0.01$ |
| | MetricBNN | $0.50 \pm 0.04$ | $0.76 \pm 0.01$ | $0.57 \pm 0.04$ | $0.95 \pm 0.01$ | $0.58 \pm 0.01$ | $0.85 \pm 0.01$ |

Table 4: Logarithmic time complexity in seconds and accuracy for the image datasets.

| | Model | MNIST | FMNIST | CIFAR10 |
|---|---|---|---|---|
| Log Time | LA GN | $3.57 \pm 0.02$ | $2.24 \pm 0.08$ | $4.79 \pm 0.16$ |
| | LA LowRank | $2.86 \pm 0.11$ | $1.67 \pm 0.06$ | $3.83 \pm 0.02$ |
| | Riem LA | $5.22 \pm 3.16$ | $5.01 \pm 4.09$ | $5.31 \pm 0.08$ |
| | Lin LA | $3.64 \pm 2.80$ | $3.67 \pm 2.96$ | $4.81 \pm 0.06$ |
| | Riem Lin LA | $3.90 \pm 3.18$ | $3.91 \pm 3.29$ | $5.50 \pm 0.09$ |
| | ENSEMBLES | $4.07 \pm 0.04$ | $4.03 \pm 0.05$ | $5.35 \pm 0.01$ |
| | SGLD | $2.97 \pm 0.05$ | $2.97 \pm 0.05$ | $4.17 \pm 0.08$ |
| | ESPRO | $2.12 \pm 0.05$ | $2.10 \pm 0.05$ | $3.93 \pm 0.10$ |
| | SWAG | $2.51 \pm 0.06$ | $2.51 \pm 0.06$ | $4.20 \pm 0.10$ |
| | MetricBNN | $3.61 \pm 0.05$ | $3.67 \pm 0.09$ | $4.22 \pm 0.10$ |
| Accuracy | LA GN | $0.19 \pm 0.02$ | $0.19 \pm 0.04$ | $0.16 \pm 0.04$ |
| | LA LowRank | $0.77 \pm 0.04$ | $0.45 \pm 0.03$ | $0.26 \pm 0.06$ |
| | Riem LA | $0.71 \pm 0.04$ | $0.38 \pm 0.03$ | $0.45 \pm 0.05$ |
| | Lin LA | $0.85 \pm 0.03$ | $0.67 \pm 0.04$ | $0.52 \pm 0.04$ |
| | Riem Lin LA | $0.92 \pm 0.02$ | $0.75 \pm 0.06$ | $0.82 \pm 0.07$ |
| | ENSEMBLES | $0.98 \pm 0.01$ | $0.87 \pm 0.02$ | $0.86 \pm 0.02$ |
| | SGLD | $0.38 \pm 0.14$ | $0.25 \pm 0.09$ | $0.85 \pm 0.10$ |
| | ESPRO | $0.96 \pm 0.02$ | $0.83 \pm 0.03$ | $0.86 \pm 0.03$ |
| | SWAG | $0.97 \pm 0.02$ | $0.77 \pm 0.02$ | $0.16 \pm 0.03$ |
| | MetricBNN | $0.97 \pm 0.03$ | $0.82 \pm 0.02$ | $0.85 \pm 0.40$ |

# D  Sensitivity of the Hyper-Parameters for MetricBNN

We additionally provide experiments on the sensitivity of the hyper-parameters of MetricBNN. Figures 8 and 9 show the difference in the sampling distribution of MetricBNN when varying the parameters $\alpha$, $N$, $T$. As expected the variance increases with an increase in the number and length of the trajectories as well as the norm of the drift term. Figures 10 and 11 provide the negative loglikelihood for the same varying parameters plus the number of hidden dimensions of the autoencoder $k$ for the posterior distribution.

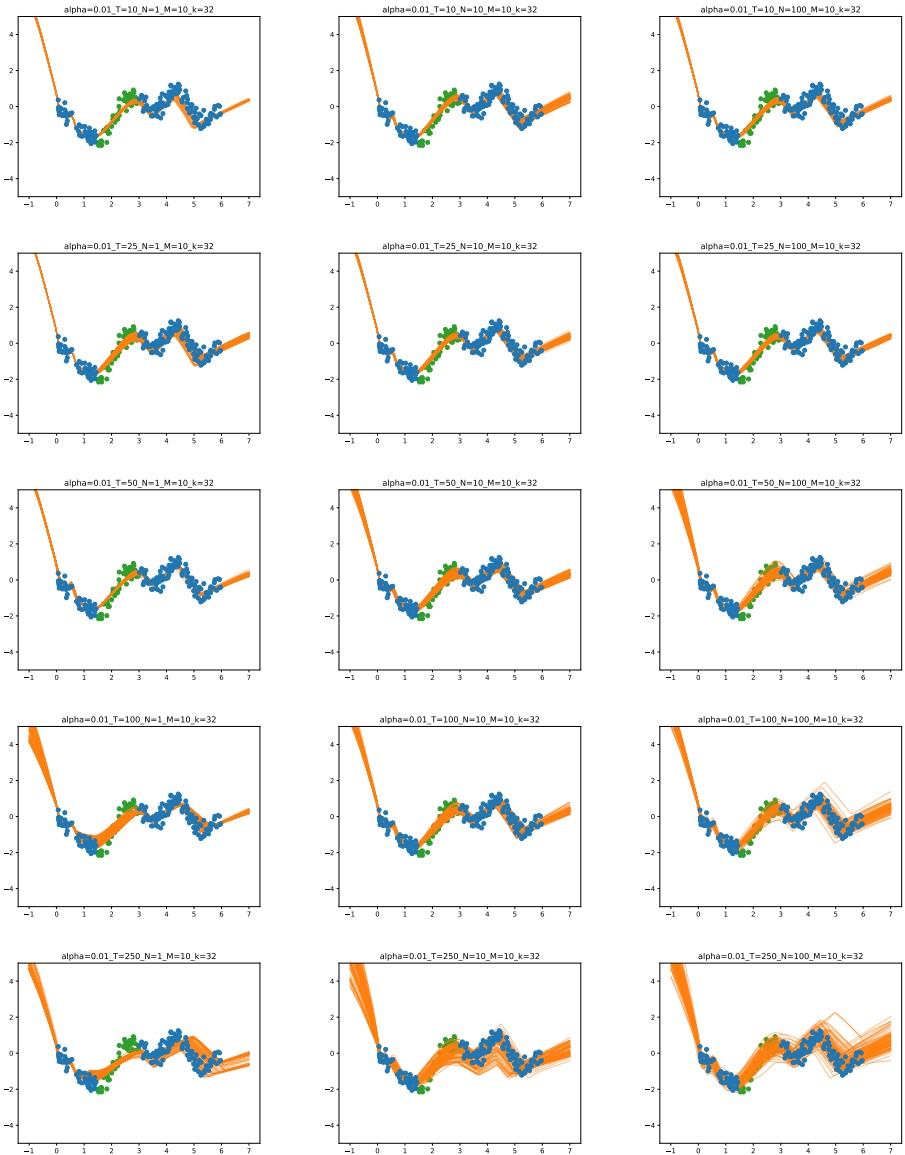

Figure 8: Samples from MetricBNN on the Toy regression problem with varying length of sampling ($T$) and number of particles ($N$).

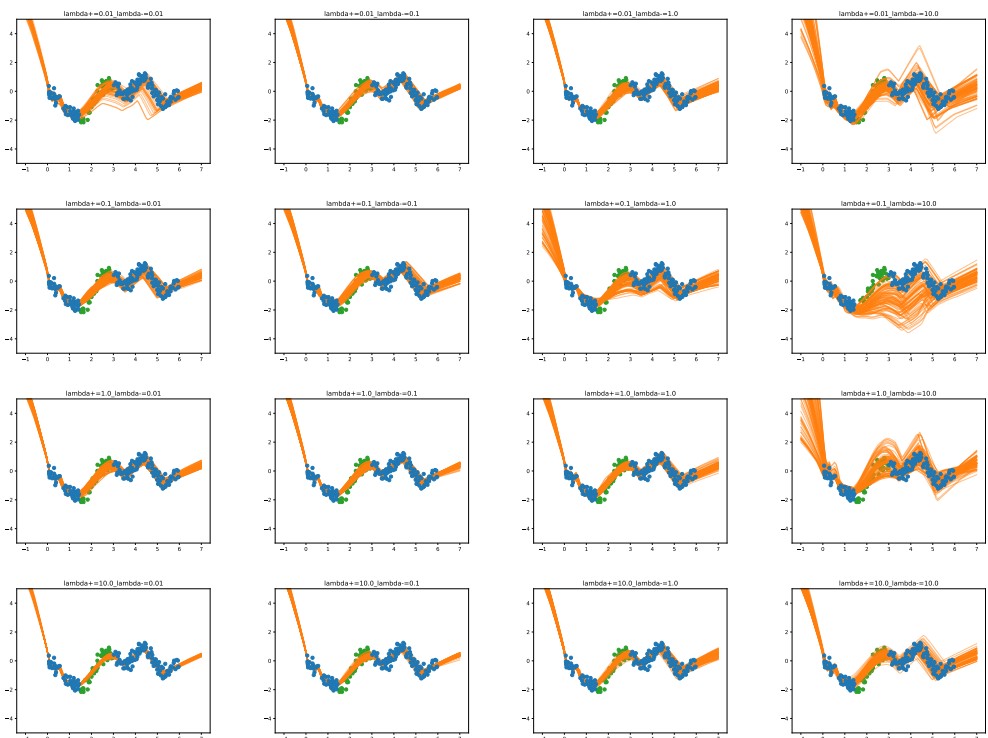

Figure 9: Samples from MetricBNN on the Toy regression problem with varying $\lambda_+$ and $\lambda_-$ for training the autoencoder ($\lambda_d$ fixed to 1).

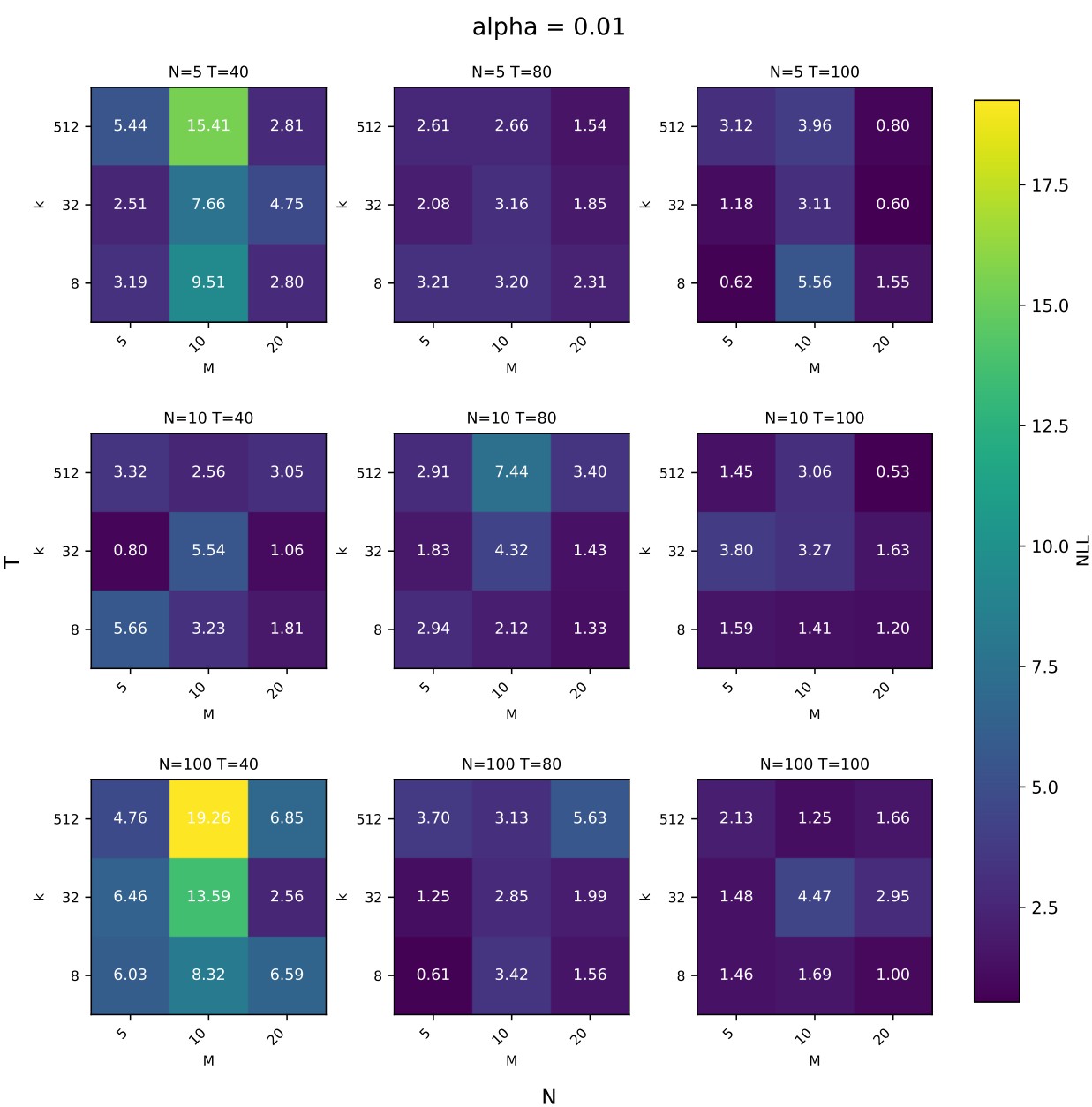

Figure 10: Negative log-likelihood on the Toy regression experiment for MetricBNN with $\alpha = 0.01$ and different values of $N$ and $T$ (different subplots) and $M$ and $k$ (within each subplot).

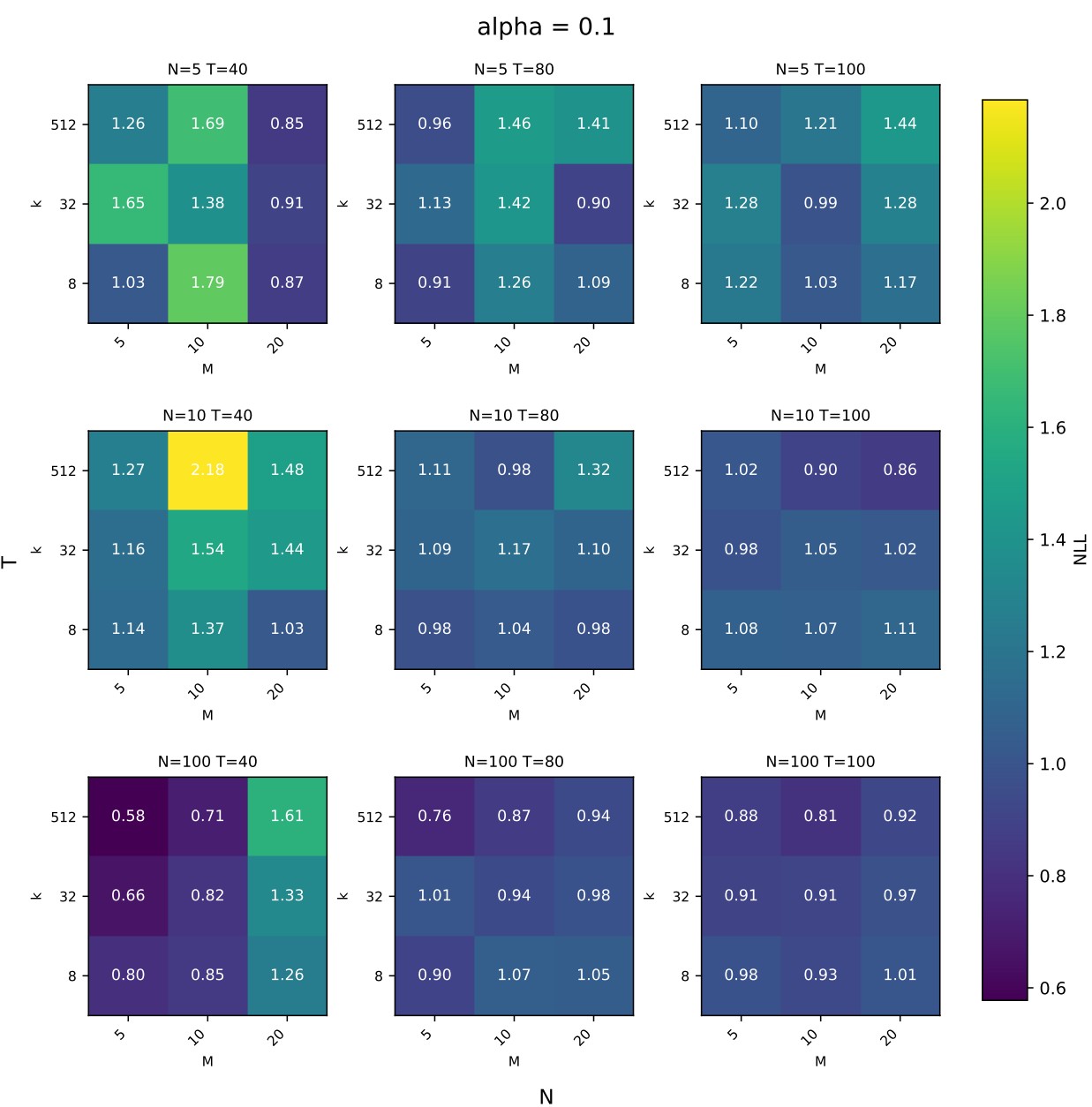

Figure 11: Negative log-likelihood on the Toy regression experiment for MetricBNN with $\alpha = 0.1$ and different values of $N$ and $T$ (different subplots) and $M$ and $k$ (within each subplot).

# E   Principal Components Analysis on Estimated Posterior

We additionally present the two principal components of the posterior samples of both MetricBNN and HMC for the Banana dataset with varying amounts of steps $T$ in Figure 12. Given enough steps, the presented method covers a large part of the parameter space.

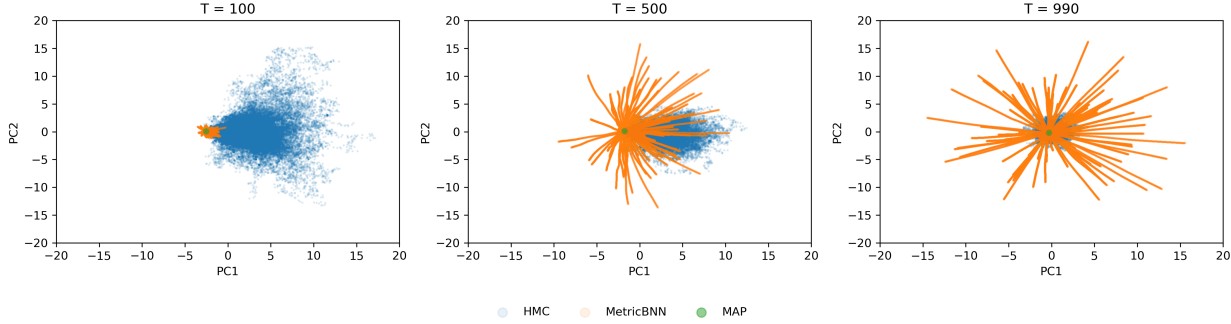

Figure 12

