# OpenReview forum: "Walking on the Fiber: A Simple Geometric Approximation for Bayesian Neural Networks"
_TMLR — Accepted by TMLR_

### Review · Reviewer_457L · 2025-04-30

**Summary Of Contributions:**

This paper proposes 'MetricBNN' a novel, sampling based approach for approximating the posterior distribution of a BNN. Like the Laplace approximation for BNNs, MetricBNN is applied post-hoc to MAP estimated NNs. MetricBNN has two components. First, a series of sample solutions near the MAP solution are found by a sequence of random perturbation + gradient update steps. Then, an AutoEncoder is fit to the sample solutions in order to create a low-dimensional latent space that can easily be sampled from. On small scale regression and classification problems, MetricBNN performs favorably compared to several Laplace approximation baselines.

**Audience:**

Yes

**Claims And Evidence:**

No

**Requested Changes:**

1. [critical] Ensure that all existing comparisons to the Laplace approximation are to variants of the Laplace approximation that are used in practice (such as those discussed in Daxberger at al. (2021).
2. [critical] Add Deep Ensembles and HMC as baselines for all existing experimental results. Deep Ensembles are a standard baseline for Bayesian Deep Learning papers as they are simple to implement, commonly used in practice, and provide a common point of reference. HMC is a highly relevant baseline for this work since it represents an alternative sampling method that practitioners might consider using.
3. [critical] Add the rotated MNIST, corrupted CIFAR, and OOD detection benchmarks from https://arxiv.org/pdf/1906.02530 and https://arxiv.org/pdf/2010.14689. These are standard benchmarks in the Bayesian Deep Learning community.
4. [critical] Add results for networks at least the size for ResNet18. This is a very small network by todays standards. MetricBNN is likely to struggle to scale to networks of this size, but the authors cannot make claims that their method is scalable when only toy sized networks are used in their results.
5. [critical] Add an ablation to show the importance of both of the components of MetricBNN. For example, what happens when you try sample directly in parameter space without the AutoEncoder component? What happens as you vary $N$ and $T$? What about the choices for $\lambda_+$, $\lambda_-$, $\lambda_d$?
6. [strengthen] Add qualitative results showing how well (or not) the samples from MetricBNN represent the posterior of the BNN. For example, compare the MetricBNN samples to those from HMC when projected into the top 2 principle components of the parameter space.
7. [critical] Add details about the size of the AutoEncoder network. Add ablations for the size of this network relative to the BNN. Add ablations for the size of the latent space.

I understand that these are big changes, but I strongly believe that this is the standard that minimum standard that the Bayesian Deep Learning community must hold itself to.

I also want to clarify that if these changes are made, I would be happy to accept this paper, regardless of the results of the experiments. I would not expect the method to be the best in all benchmarks and against all baselines. However, it is important for the paper to accurately characterize the method so that practitioners can make and informed choice about whether to use the method and so that future work in the field can be built on strong foundations. I think that the proposed method is interesting and could have several strengths, but unfortunately it isn't possible to know that with the current experimental evaluation.

**Strengths And Weaknesses:**

### Strengths

* The paper is clearly written.
* The proposed method is simple yet interesting.

## Weaknesses

* **Framing** – While the proposed method is similar to the Laplace approximation in that both methods can be applied post-hoc, I found the constant comparisons with the Laplace approximation to be unnecessary/distracting. Of course, it is normal to compare a new method with relevant baselines, but it is usually a good idea to (a) compare with a range of relevant methods, and (b) motivate the method on its own strengths rather than another methods weaknesses. Furthermore, many of the comparisons (in both the text and results) were made with a very specific instance of the Laplace approximations (e.g., not linearised, and/or without any common further scalability approximations such as last-layer or block-diagonal structure). Ultimately, this makes the comparisons somewhat meaningless (and misleading) since these are the methods that are used in practice).
* **Experimental Evaluation** – Unfortunately, the experimental evaluation for this paper does not meet the par for publication in my opinion. There are several areas that I feel are lacking. (1) Comparison with relevant baselines such as deep ensembles and HMC are missing. (2) Evaluations on NNs of even modestly large sizes by todays standard. (3) Evaluations on standard uncertainty estimation benchmarks such as OOD detection and dataset shift. (4) Ablations for the different components/modeling choices in MetricBNN. (5) Qualitative experiments that help to build intuition for how MetricBNNs works (and where it might not work). I provide some concrete suggestions for improving these issues below.

---

> ### Author Response · Authors · 2025-05-31
> **Response to Reviewer 457L**
>
> We thank the Reviewer for the feedback and the suggested improvements. Below some comments on the requested changes.
>
> **[critical] Ensure that all existing comparisons to the Laplace approximation are to variants of the Laplace approximation that are used in practice (such as those discussed in Daxberger at al. (2021).**
> We have updated the document. For the Laplace approximation we use the Gauss-Newton approximation, the Low-Rank approximation and the linear adaptation.
>
> **[critical] Add Deep Ensembles and HMC as baselines for all existing experimental results. Deep Ensembles are a standard baseline for Bayesian Deep Learning papers as they are simple to implement, commonly used in practice, and provide a common point of reference. HMC is a highly relevant baseline for this work since it represents an alternative sampling method that practitioners might consider using.**
> We have updated the document with additional set of baselines and experiments.
>
> **[critical] Add the rotated MNIST, corrupted CIFAR, and OOD detection benchmarks from https://arxiv.org/pdf/1906.02530 and https://arxiv.org/pdf/2010.14689. These are standard benchmarks in the Bayesian Deep Learning community.**
> We have updated the document with additional set of baselines and experiments.
>
> **[critical] Add results for networks at least the size for ResNet18. This is a very small network by todays standards. MetricBNN is likely to struggle to scale to networks of this size, but the authors cannot make claims that their method is scalable when only toy sized networks are used in their results.**
> We have added an additional classification experiment on CIFAR10 with a ResNet18 architecture.
>
> **[critical] Add an ablation to show the importance of both of the components of MetricBNN. For example, what happens when you try sample directly in parameter space without the AutoEncoder component? What happens as you vary N and T? What about the choices for $\lambda_+$, $\lambda_-$, $\lambda_d$?**
> We have included additional results in the Appendix showing the effect of different parameters of MetricBNN. Figure 3 shows what happens when naively approximating the distribution of the parameters found with a Gaussian (SVD approximation).
>
> **[strengthen] Add qualitative results showing how well (or not) the samples from MetricBNN represent the posterior of the BNN. For example, compare the MetricBNN samples to those from HMC when projected into the top 2 principle components of the parameter space.**
> We have included additional results in the Appendix.
>
> **[critical] Add details about the size of the AutoEncoder network. Add ablations for the size of this network relative to the BNN. Add ablations for the size of the latent space.**
> We have included additional results in the Appendix.

---

> > ### Comment · Reviewer_457L · 2025-06-18
> > **Minor suggestion**
> >
> > Thank you for taking the time to address my comments so thoroughly. I am now satisfied that this paper meets the TMLR acceptance criteria.
> >
> > I have one minor suggestion which is to replace the large tables (1, 2) with bar plots to make it easier to compare the methods. Similarly, while figure 7 is fantastic for getting a detailed picture of the Corrupted CIFAR results, I would move it to the appendix, and use a summarized plot that averages over corruptions in the main text. Also, it wasn't clear to me what strength of corruption was applied? Usually there are 5 different strengths, and one can make a plot of performance vs corruption strength much like the Rotated MNIST setting.

---

### Review · Reviewer_hmoS · 2025-05-01

**Summary Of Contributions:**

This paper introduces MetricBNN, a two-stage, post-hoc Bayesian procedure for neural networks. Stage 1 draws a cloud of weight vectors by repeatedly adding small random “drift” perturbations to the maximum-a-posteriori (MAP) solution and then re-optimizing for a few gradient steps, exploiting the empirical observation that good solutions lie on a low-dimensional, connected manifold in weight space. Stage 2 trains an auto-encoder on those samples so that, in the learned latent space, entire trajectories become nearly linear; new posterior samples are produced by simply interpolating between the latent MAP point and the endpoint of a chosen trajectory and decoding back to weights. Across toy regression/classification tasks, six UCI datasets, MNIST and Fashion-MNIST, MetricBNN delivers negative-log-likelihoods on par with, or better than, Laplace and Riemannian-Laplace refinements while requiring markedly less Hessian-related computation and scaling almost flatly with network depth.

**Audience:**

Yes

**Broader Impact Concerns:**

None.

**Claims And Evidence:**

Yes

**Requested Changes:**

Can you provide a sensitivity analysis for α, N, T, M and latent dimension k, and clarify how many samples are really needed to train a useful auto-encoder?

What is the role of 1/T in Eq. (7)?

The drift-and-refine procedure seems local; how do you ensure that distinct posterior modes (if any) are visited rather than merely thickening a single basin?

What prevents the decoder from mapping interpolations to weights with poor training loss, especially as interpolation length grows? Have you measured the actual loss of decoded samples before using them at test time?

Could you compare against SWAG or SGLD on at least one dataset to illustrate where MetricBNN sits on the accuracy–cost frontier?

Figure 5 should include a legend instead of explaining what each color represents. There seems to be a difference in the trend where the proposed sampling method shows decreasing NLL for larger networks/time, while both Laplace approximation methods see an increase in NLL. Why does sampling on a low-dimensional manifold fix the overfitting?

**Strengths And Weaknesses:**

[Strengths]

Elegant use of geometry: By “walking on the fiber” instead of relying on Hessian curvature, the method sidesteps the positive-definiteness issues and cubic costs that plague Laplace in over-parameterized nets.

Fast sampling after training: Once the auto-encoder is fitted, drawing a posterior sample is reduced to one latent-space interpolation and one forward pass through the decoder, enabling near-instant Monte-Carlo predictions.

Versatility and minimal intrusiveness: The algorithm is completely post-hoc—no change to the original optimizer or loss—and is demonstrated on fully-connected, convolutional and shallow networks, as well as both regression and classification problems.

Competitive empirical performance: MetricBNN matches or outperforms four Laplace-based baselines on NLL while keeping wall-time low, and its uncertainty bands on toy problems visually capture data gaps that Laplace approximation misses.

[Weaknesses]

Limited theoretical grounding: The paper assumes, but does not rigorously justify, that solution sets form one-dimensional fibers and that linear interpolation in latent space remains within a high-probability posterior region.

Modest experimental scale: All image experiments use small-footprint CNNs on MNIST-like data; there is no evidence the approach remains tractable for modern large language/vision models. Even adding an experiment on small transformer model would greatly enhance the generalizability of their conclusions.

Hyper-parameter sensitivity: Performance may hinge on drift scale α, number of particles N, sampling steps T and auto-encoder dimension k, yet only a single setting is reported and no robustness study appears.

Comparative breadth: Baselines omit state-of-the-art stochastic methods such as SGLD, SGHMC, SWAG and deep ensembles, so it is unclear how MetricBNN fares against the broader Bayesian-deep-learning toolkit.

Evaluation metrics: The study relies almost exclusively on NLL; calibration (ECE), out-of-distribution detection, or posterior-predictive checks are not reported, leaving uncertainty quality only partially assessed.

---

> ### Author Response · Authors · 2025-05-31
> **Response to Reviewer hmoS**
>
> We thank the Reviewer for the feedback and the suggested improvements. Below some comments on the requested changes.
>
> **Can you provide a sensitivity analysis for $\alpha$, N, T, M and latent dimension k, and clarify how many samples are really needed to train a useful auto-encoder?**
> We have included additional results in the Appendix.
>
> **What is the role of 1/T in Eq. (7)?**
> The term 1/T is not strictly needed. Two subsequent set of parameters need, in fact, to be placed within some radius to eachother in the learned latent space. Using 1/T ensures that the entire trajectory of sampled parameters is within a Euclidean distance of 1 which makes the log oof the negative term strong enough.
>
> **The drift-and-refine procedure seems local; how do you ensure that distinct posterior modes (if any) are visited rather than merely thickening a single basin?**
> The reviewer is correct in pointing out the inherent locality of the method. This work is based on the assumption that the space of solutions is connected in the parameter space. This is an assumption that other methods make implicitly as well, e.g., any Hessian-based method.
>
> **What prevents the decoder from mapping interpolations to weights with poor training loss, especially as interpolation length grows? Have you measured the actual loss of decoded samples before using them at test time?**
> Each particle is first nudged by a tiny random drift and then pulled back with M gradient-descent refinements (Eq. 4-5). This guarantees that every checkpoint we keep remains a solution of low training loss . Because consecutive checkpoints differ only by a small perturbation, the straight line that connects them in weight space is empirically still inside the same basin. The positive loss of the decoder keeps successive checkpoints one unit apart so local distances are preserved. The negative term, instead, “uncoils’’ the trajectory, encouraging it to become a straight segment in latent space. These complementary forces make movement in Z continuous and well-behaved: linear interpolation simply walks along that straight segment, so decoding any intermediate point still yields weights inside the original solution basin.
>
> **Could you compare against SWAG or SGLD on at least one dataset to illustrate where MetricBNN sits on the accuracy–cost frontier?**
> We have updated the document with additional set of baselines and experiments.
>
> **Figure 5 should include a legend instead of explaining what each color represents. There seems to be a difference in the trend where the proposed sampling method shows decreasing NLL for larger networks/time, while both Laplace approximation methods see an increase in NLL. Why does sampling on a low-dimensional manifold fix the overfitting?**
> We have updated the Figure. Hessian-based methods scale poorly with the dimensionality of the parameter space. In terms of cost this is because the Hessian is quadratic in complexity. In terms of NLL this is because the set of solutions in parameter space become more non-linear and a simple covariance fails in correctly capturing it. On the other hand, our proposed method’s complexity mostly depends on the number of samples collected which correlates with the intrinsic dimensionality of the solution space.

---

> > ### Comment · Reviewer_hmoS · 2025-06-18
> > **Thank you for addressing all of my concerns**
> >
> > In their revision, the authors have addressed all of my initial concerns. I have no objections against accepting this paper.

---

### Review · Reviewer_rEb8 · 2025-05-05

**Summary Of Contributions:**

This paper proposes a sample then optimize approach for Bayesian neural networks. Specifically, a neural network is trained to completion, then noise is added to corrupt the solution, after which a few gradient steps are taken. When an explicit distribution, rather than samples, are desired, the authors train an autoencoder on a low dimensional space as a sort of hypernetwork to generate new sets of weights.
Experiments are performed on a toy classification task, MNIST classification, and UCI regression.

**Audience:**

Yes

**Claims And Evidence:**

No

**Requested Changes:**

Please compare the NLL and log time to some sampling based approaches, e.g. SGMCMC or something analogous. Similarly, a comparison with the low dimensional approach in Benton et al, ’21 would be quite useful as it’s similarly inspired by the mode connectivity work of Garipov et al, ’18.

Ensembles (Lakshimarayan et al, ’16) are also a very good, missing baseline in the two quantitative experiments.

Figure 5: What parameter sizes are the networks here?

Is this the time taken to compute the NLL? With how many samples?

Could accuracy metrics be reported for Table 2?

Could a larger experiment, even CIFAR 10 with conv nets, be performed to test the practical scalability of the approach?

How many samples are used for training the metric BNN autoencoder, and what is its typical computational cost?

**Strengths And Weaknesses:**

**Strengths:**

The writing is overall pretty clear especially in the methodological description. Good job.

There are several understanding experiments towards the practical implications of this type of posterior approximation.

**Weaknesses:**

There’s unfortunately a fair amount of missing work here, and a few incorrect claims:

- Pg 2: “Sampling-based methods such as stochastic gradient langevin dynamics … [are computationally costly] and impractical for large-scale neural networks.” Yes, HMC is extremely computationally costly (see Wilson & Izmailov, 20 and Izmailov et al, ’21). However, stochastic gradient MCMC is not actually much more costly than standard training (each gradient step is the same cost as standard training), its just the inference costs can be much higher due to the sample collection required – see Welling & Teh, ’11, Zhang et al, ’20.

- Figure 4: Qualitatively, the SVD approximation here displays more of the behavior of a neural network that we would expect – it rapidly becomes unconfident off the data manifold. By comparison, the metricBNN distribution stays very confident in most places, even more so than the “sampling distribution”.

- The overall idea is very similar to randomize – then – optimize in the applied mathematics literature, see Bardsley et al, 14, just with direct optimization instead of sampling thereafter, and to Pearce et al ‘20’s direct sampling around the posterior, except again with direct optimization instead of an injected prior. The idea is also pretty well known in the RL literature for neural linear models typically.

- The autoencoder piece is novel; however, some of the follow up works to Garipov et al, ’18 were able to successfully get low dimensional distributions in the weight space of the neural networks without the need for autoencoders. See for example, Benton et al, ’21 and Wortsman et al, ’21 – the former is more explicitly probabilistic; however, both have Bayesian themes.

- The resemblance to Bayesian hypernetworks (Kruger et al, ’18) should also be discussed as that approach directly trains the autoencoder from scratch to generate weight samples of the NN.

References:
“cyclical stochastic gradient mcmc for Bayesian deep learning,” zhang et al, iclr 2020.

“what are Bayesian neural network posteriors really like,” izmailov et al, icml 2021.

“Bayesian deep learning and a probabilistic perspective on generalization,” Wilson & izmailov, neurips, 2020.

“Bayesian learning via stochastic gradient langevin dynamics,” welling & teh, icml, 2011.

“uncertainty in neural networks: approximately Bayesian ensembling,” pearce et al, aistats, 2020.

“Randomize-Then-Optimize: A Method for Sampling from Posterior Distributions in Nonlinear Inverse Problems”, Bardsley et al, SIAM Journal on Scientific Computing, 2014.

“loss surface simplexes for mode connecting volumes and fast ensembling”, Benton et al, icml, 2021.

“Learning neural network subspaces,” Wortsman et al, 2021.

“Simple and Scalable Predictive Uncertainty Estimation using Deep Ensembles,” Lakshminarayanan et al, 2016

“Bayesian hypernetworks,” Kruger et al, 2018. https://arxiv.org/pdf/1710.04759

---

> ### Author Response · Authors · 2025-05-31
> **Response to Reviewer rEb8**
>
> We thank the Reviewer for the feedback and the suggested improvements. Below some comments on the requested changes.
>
> **Compare the NLL and log time to SGMCMC or something analogous, Compare with Benton et al, ’21**
> We have updated the document with additional set of baselines and experiments.
>
> **Ensembles (Lakshimarayan et al, ’16) are also a very good, missing baseline in the two quantitative experiments.**
> We have updated the document with additional set of baselines and experiments.
>
> **Figure 5: What parameter sizes are the networks here?**
> We have expanded the figure with more baselines and more layers. The networks vary from 1 layer up to 9 layers, hidden layers all have 64 neurons and a Relu activation function. We have included these details in the Appendix.
>
> **Is this the time taken to compute the NLL? With how many samples?**
> Yes, we use 100 parameters’ samples to approximate the posterior. We have included this in the Appendix.
>
> **Could accuracy metrics be reported for Table 2?**
> We have included additional results in the Appendix.
>
> **Could a larger experiment, even CIFAR10 with conv nets, be performed to test the practical scalability of the approach?**
> We have included an experiment  on CIFAR10 with a ResNet18 architecture in Table 2.
>
> **How many samples are used for training the metric BNN autoencoder, and what is its typical computational cost?**
> To train the autoencoder we use the samples collected during the sampling steps, i.e. N x T. The reported numbers on the time include both the sampling step as well as the training of the autoencoder.

---

> > ### Comment · Reviewer_rEb8 · 2025-06-07
> > **thanks for the updates**
> >
> > a couple of quick comments on the updated paper:
> >
> > - figure 5: "ensembles" not "ensambles". this plot is now a bit illegible, and i'm frankly confused as to why a single ensemble has that high of a NLL. misnaming is in table 1 as well.
> >
> > - the laplace approximations seem to perform much worse than in their original papers, which is also confusing to me

---

> > > ### Author Response · Authors · 2025-06-10
> > >
> > > We thank the reviewer once again for providing feedback to the updated manuscript. Here are some replies to the comments:
> > >
> > > **figure 5: "ensembles" not "ensambles". this plot is now a bit illegible, and i'm frankly confused as to why a single ensemble has that high of a NLL. misnaming is in table 1 as well.**
> > >
> > > We thank the reviewer for spotting the misspelling; “ensemble” is now used consistently in Table 1, Figure 5, and throughout the text.To improve legibility, Figure 5 has been regenerated. Regarding the unexpectedly high NLL for the 1-hidden-layer ensemble, this architecture is under-parameterized for the dataset’s size and dimensionality, causing the independently initialized networks to collapse onto essentially the same MAP solution—yielding accurate predictive means but severely underestimated variances and, therefore, over-confident predictions that inflate the NLL.
> > >
> > > **the laplace approximations seem to perform much worse than in their original papers, which is also confusing to me**
> > >
> > > We acknowledge that results might differ from the original papers. The Laplace baselines were implemented with the open-source Laplace library (https://github.com/aleximmer/Laplace). These geometric methods are notably sensitive to model size and parameter count, and their performance is further tied to the quality of the maximum-a-posteriori (MAP) network on which the approximation is centered. Differences in network capacity or MAP training conditions therefore translate into the lower scores observed in our experiments.

---

### Decision · Action_Editor_Yvxe · 2025-06-24

**Recommendation:** Accept with minor revision

**Additional Comments:**

The following minor changes are requested for full acceptance of the paper:

- A discussion of the performance of the Laplace method in these experiments compared to other published papers using Laplace
- A more extended discussion of the new empirical results that have been added in the revision
- A more extended discussion of how this approach differs from different types of hypernetworks
- Fixing the typo "ensambles" -> "ensembles" throughout the text and figures (e.g., Fig. 2, Fig. 4)

Once these changes have been made in the camera-ready version, the paper will be ready for publication in TMLR.

**Audience:**

Yes

**Audience Explanation:**

The reviewers agree that at least some subset of the TMLR audience will find this paper interesting.

**Claims And Evidence:**

Yes

**Claims Explanation:**

The reviewers found that the revision has addressed most of their concerns and that most claims are now supported by clear evidence. A few minor issues remain, which are outlined below.